# SUMO-Based Regulation of Nuclear Positioning to Spatially Regulate Homologous Recombination Activities at Replication Stress Sites

**DOI:** 10.3390/genes12122010

**Published:** 2021-12-17

**Authors:** Kamila Schirmeisen, Sarah A. E. Lambert, Karol Kramarz

**Affiliations:** 1Institut Curie, Université PSL, CNRS UMR3348, 91400 Orsay, France; kamila.schirmeisen@curie.fr; 2Ligue Nationale Contre le Cancer (Équipe Labélisée), Université Paris-Saclay, CNRS UMR3348, 91400 Orsay, France; 3Academic Excellence Hub—Research Centre for DNA Repair and Replication, University of Wrocław, 50-328 Wrocław, Poland

**Keywords:** DNA, replication stress, SUMO, genome stability, homologous recombination, nuclear pore complex, chromatin mobility, yeast

## Abstract

DNA lesions have properties that allow them to escape their nuclear compartment to achieve DNA repair in another one. Recent studies uncovered that the replication fork, when its progression is impaired, exhibits increased mobility when changing nuclear positioning and anchors to nuclear pore complexes, where specific types of homologous recombination pathways take place. In yeast models, increasing evidence points out that nuclear positioning is regulated by small ubiquitin-like modifier (SUMO) metabolism, which is pivotal to maintaining genome integrity at sites of replication stress. Here, we review how SUMO-based pathways are instrumental to spatially segregate the subsequent steps of homologous recombination during replication fork restart. In particular, we discussed how routing towards nuclear pore complex anchorage allows distinct homologous recombination pathways to take place at halted replication forks.

## 1. Replication Stressed Forks and Homologous Recombination

In an average human life span, each individual copies approximatively 2 × 10^16^ m of DNA, representing 130,000 times the distance between the earth and the sun. DNA replication is therefore a fundamental process necessary for cell division, organism development, tissue homeostasis, and cell renewal. Genome duplication occurs during S-phase, and the associated DNA synthesis is overall highly accurate. Nonetheless, many endogenous and exogenous factors can challenge the process of DNA replication, a phenomenon that is referred to as replication stress. Replication stress can be defined as any event that alters the rate of DNA replication. This includes the deceleration of replication fork progression, a well-recognized feature of replication stress, as consequence of a myriad of fork obstacles [1]. The rate of fork progression can be affected globally upon treatment with chemotherapeutics drugs targeting DNA replication, oncogene activation, or inherited mutations that impair DNA replication [2,3]. In addition, during each round of DNA replication, a myriad of fork obstacles have the potential to hinder DNA synthesis, making particular genomic regions difficult to replicate, such as telomeres, centromeres, and sites of transcription–replication conflicts [3]. Replication blocks can result in the slowdown, stalling, or collapse of the replisome. Stressed replication forks are fragile DNA structures prone to DNA breakage leading to mutation and gross chromosomal rearrangements. Beyond the challenge of maintaining genome stability, replication stress induces a cascade of cellular processes, such as inflammation, senescence, aging, and cell death affecting cell fate and identity [2,4]. Therefore, replication stress is an underlying cause of many human diseases, including cancer, in-born developmental defects, neurological disorders, and accelerated aging. For example, the cancer risk of a given tissue is mathematically linked with the number of stem cell divisions, and cancer development and aggressiveness is associated with intrinsic replication stress [5,6]. The molecular processes that govern the accuracy of genome duplication upon physiological or pathological replication stress have been under intense research, at both the basic and clinical level, with the aim to target novel pathways to cure diseases.

The maintenance of genome stability upon replication stress relies on the completion of DNA replication and numerous replication fork repair pathways that have evolved with increasing genome sizes through evolution [7]. Among these pathways, homologous recombination (HR) is particularly active in protecting, repairing, and restarting stressed replication forks [8,9]. HR repairs broken replication forks through a mechanism called break-induced replication (BIR) and ensures replication resumption at double strand break-free (DSB-free) arrested forks through template switching or a mechanism called recombination-dependent replication (RDR) [10,11,12]. This last pathway is initiated by Rad51-coated single-stranded DNA gaps formed through the well-controlled degradation of newly replicated strands [13,14,15,16]. Both BIR and RDR are associated with mutagenic DNA synthesis, which distinguishes a restarted fork from a replication origin-born fork [12]. This feature might be particularly harmful when the fork arrests within repeated sequences. Akin to how nuclear positioning impacts the way a double-strand break (DSB) is repaired, recent advances support the hypothesis that molecular transactions engaged at arrested forks depend on nuclear positioning, in which SUMO-based mechanisms are critical. Here, we review how the spatially segregated SUMO metabolism in yeast nuclei regulates the distinct steps of HR-mediated fork restart and the relevance of this in human cells.

## 2. Replication Stress Sites Move to the Nuclear Periphery

Eukaryotic genomes are 3D folded in a highly compartmentalized nucleus that has a distinct chromatin environment and DNA repair capacity [17]. In the early 2000s, it was discovered that damaged chromatin exhibit increased mobility to allow DNA damages to shift away from their compartment to another one to complete DNA repair [18,19,20]. This includes DSBs occurring within heterochromatin in *Drosophila*, yeast, human nucleolus, and mouse peri-centromeres that escape their compartment to achieve DSB repair through HR [21,22,23,24,25]. This led to the concept that a given chromatin environment is refractory to DNA repair processes and that DNA repair machineries are spatially segregated [17,26]. In budding yeast, difficult to repair DSBs (*i.e.,* in the absence of donor template for HR repair) at unique sequence are mobilized to the nuclear periphery to anchor to components of the nuclear pore complex (NPC) or the nuclear envelope to achieve DNA repair by salvage pathways [20,27,28,29]. Eroded telomeres (i.e., in the absence of telomerase), which mimic one-ended DSB, also anchor to NPCs to ensure the maintenance of telomere length by HR (referred to as type II survivors) [30,31]. The necessity of changing nuclear compartment for NPC anchorage has been extended to halted replication forks. In yeast models, forks stalling within telomere repeats, forks stalled by tri-nucleotides repeats or DNA-bound proteins, and collapsed forks relocate to the nuclear periphery for NPC anchorage [20,32,33,34]. In human cells, forks stalled upon the inhibition of DNA polymerases exhibit relocation to the nuclear periphery, and replication stress at telomeres leads to telomeres’ association with NPCs [35,36]. Preventing relocation results in chromosome breaks, delayed replication restarts, and abnormal mitotic chromosome segregation including micronuclei formation. The directed mobility of damaged chromatin to its relocation at the nuclear periphery requires nuclear forces provided by microtubules and nuclear filamentous actin, a subject recently reviewed in [37]. Of interest in this review, SUMO-based mechanisms are central to NPC anchorage and for the orchestration of the subsequent steps of DSB repair by HR [20,21,27,31,32,33,38]. Recent studies indicate that the anchorage of replication stress sites to NPCs is controlled by SUMO metabolism for a tight regulation of HR activity. In fission yeast, a novel spatial regulation of RDR was proposed based on two sub-pathways inside the nucleus: one that occurs within the nucleoplasm and one that involves NPC anchorage [33]. This routing is regulated by SUMOylation and has a distinct outcome on the efficiency of RDR and the maintenance of fork integrity.

A long-standing question was if DSB formation was a prerequisite for NPC anchorage. Because collapsed forks are prone to breakage, it could not be excluded that forks arrested by secondary DNA structures or at telomere repeats undergo fork breakage before NPC anchorage. In fission yeast, a site-specific replication fork barrier (called *RTS*1-RFB) allows the polar block of a single replisome by a DNA-bound protein complex [39]. Forks arrested by the RFB are bound by HR factors, including the recombinase Rad51 and its loading mediator Rad52, independently of DSB formation. Instead, the binding of HR factors requires the controlled degradation of newly replicated strands by nucleases (i.e., MRN-Ctp1, Exo1) to generate a single-stranded DNA gap [16,40,41]. In such a system, the active RFB anchors to the NPC for the time necessary for HR to restart the arrested fork [33,42,43], supporting the hypothesis that DSB is not a prerequisite for the anchorage of replication stress sites.

## 3. SUMOylation in DNA Repair

SUMO (small ubiquitin-like modifier) is an essential particle present in all eukaryotic cells that triggers post-translational modifications (PTMs) (Figure 1). Akin to ubiquitin, SUMO is covalently attached to target proteins. SUMOylation affects the activity, localization, and stability of modified proteins. All SUMO particles are expressed as immature precursors, which must be cleaved at the C-terminus by sentrin/SUMO-specific proteases (SENPs) to expose two glycine residues essential for further conjugation [44]. Subsequently, after activation and transesterification by the E1 enzyme, SUMO is transferred onto target protein by the joint action of the E2 conjugating enzyme and a limited set of E3 SUMO ligases. Despite its great importance for cell fitness and survival, SUMOylation is not an abundant PTM, in contrast to ubiquitination. SUMO might be attached to its targets via a single acceptor lysine as a monomer, thus generating monoSUMOylation (Figure 1). If a monoSUMO particle is covalently attached to several lysines of a given substrate, it is referred to as multiSUMOylation, a type of polySUMOylation. An interesting feature of SUMO is its ability to form polymeric chains by attachment of the SUMO particle to the internal lysines of the initial SUMO particle. We will refer to this last type of modifications as a SUMO chain, another type of polySUMOylation [45]. In yeast models, SUMO is encoded by a single gene (*Saccharomyces cerevisiae SMT3* and *Schizosaccharomyces pombe pmt3^+^*), whereas higher eukaryotes express a few conjugatable SUMO paralogs (SUMO1-5) [46,47,48,49].

Pioneering studies in yeasts have revealed that monoSUMOylation plays important roles in DNA repair with numerous DNA repairs factors being SUMOylated to regulate their activity and localization, including HR factors [50,51,52]. Furthermore, studies conducted on higher eukaryotes have also described numerous SUMO targets among DNA repair proteins. For instance, SUMOylation of human CtIP (*S. pombe* Ctp1, *S. cerevisiae* Sae2) favors DNA end resection at DSBs and the protection of replication forks [53,54]. The analysis of proteins associated with nascent DNA has revealed that several components of the replisome are SUMOylated in human cells [55]. This includes DNA polymerases, the MCM complex, PCNA, and RPA [56,57,58]. Replication stress is broadly connected to an increased level of SUMOylation for many of these factors, a phenomenon called SUMO stress response (SSR), which plays key roles in preserving genome stability upon perturbed replication conditions. For instance, in budding yeast, monoSUMOylation was shown to protect damaged forks through the accumulation of Rad51-dependent recombination DNA structures [59,60]. Furthermore, in budding yeast, RPA becomes polySUMOylated during replicative senescence. At stalled replisomes several factors of replication restart machineries undergo SUMOylation, such as Mre11, Ku, Sgs1, and Rad52 [61,62]. Nonetheless, the repertoire of SUMOylated factors at replication forks in response to a distinct type of replication stress largely remains to be established.

SUMO chains are detectable in all eukaryotic organisms, especially in response to replication stress. Although SUMO chains help to target proteins for degradation by the proteasome, their potential contributions in regulating DNA repair or replication processes remain largely unfathomed [45,63]. Moreover, SUMOylation can act as a double-edged sword in sustaining genome stability; both ineffective SUMOylation and the accumulation of SUMO chains make cells sensitive to DNA damage and replication stress [64]. Any dysregulation in the SUMO level can be deleterious for cells’ survival and influence DNA repair capacities, putting SUMO metabolism under tight regulation [65,66].

In budding yeast, SUMOylation is catalyzed by three E3 SUMO ligases (Table 1). The activity of the two paralogs Siz1 and Siz2 (human PIAS1-4, *S. pombe* Pli1) is responsible for bulk SUMOylation in *S. cerevisiae* cells [67]. The third E3 SUMO ligase Mms21 (human MMS21, *S. pombe* Nse2) has fewer substrates and mainly catalyzes monoSUMOylation. Mms21 is a part of the Smc5-6 complex and is critical for DNA repair and cell survival [38]. Similarly, the *S. pombe* Mms21 homologue, Nse2, is also part of the Smc5-6 complex and mainly catalyzes monoSUMOylation, which is critical for the maintenance of chromosome integrity [64]. Therefore, the lack of Nse2 is lethal, and the mutation of the catalytic RING domain leads to severe sickness [50,68]. In fission yeast, Pli1, which triggers the formation of both monoSUMOylation and SUMO chains, conducts the bulk SUMOylation. The mutation of *pli1+* does not lead to cellular sensitivity to DNA-damaging agents, in contrast to *nse2* defects [50]. This suggest an apparent division of labor between distinct E3 SUMO ligases, but the underlying mechanisms are currently not understood.

The action of E3 SUMO ligases is antagonized by SENP SUMO proteases (Ulp1 and Ulp2 in budding and fission yeast and six SENPs in humans, see Table 1) that can directly remove SUMOylation from target proteins. The activity of SENP SUMO proteases is spatially segregated in the nucleus in most organisms. Budding yeast Ulp1 is localized at the nuclear periphery through interactions with the Y-complex of the NPC (Nup84) and the nuclear basket (Nup60–Mlp1/2), whereas Ulp2 is located in the nucleoplasm [20,74]. Importantly, Ulp1 cleaves the SUMO precursor to make it prone to conjugation with the E1 enzyme. The mutation of *ULP1* is inviable in budding yeast and in *S. pombe*, it leads to extreme sickness together with a global decrease in SUMO levels because of the defect in the SUMO conjugation cycle [50]. Cells devoid of Ulp2 exhibit poor growth in both yeast models and accumulation of high-molecular-weight (HMW) SUMO conjugates, highlighting the distinct roles of Ulp1 and Ulp2 in SUMO regulation. In human cells, SENP1, SENP2, SENP3, and SENP5 are evolutionary related to yeast Ulp1, whereas SENP6 and SENP7 are derived from Ulp2 [75]. Among this group, SENP1 and SENP2 are enriched at the nuclear periphery, and both are required for the maturation of SUMO precursors [76].

PolySUMOylated proteins are recognized and bound by specific enzymes called SUMO-targeted ubiquitin ligases (STUbLs) that transfer ubiquitin onto SUMO for proteasomal degradation to modulate nuclear localization or activity (Table 1) [77]. In budding yeast, two STUbLs have been reported so far: the heterodimer Slx5-Slx8 and the large protein Uls1. In *S. pombe*, two distinct STUbL complexes are formed by the interaction between Slx8 and either Rfp1 or Rfp2 proteins. Human cells contain RNF4 and RNF111 enzymes exhibiting STUbL activities. In general, STUbLs are enzymes containing a RING domain characteristic of E3 ubiquitin ligases and several SUMO-interacting motifs (SIMs), that enable interactions with SUMOylated proteins [78]. Defects in STUbLs activity leads to a drastic increase in HMW-SUMO conjugates in cells [79,80]. Interestingly, budding yeast Ulp2 and human SENP6 were found to antagonize STUbLs by restraining SUMO chains’ generation in the nucleoplasm [81,82,83]. Beyond triggering protein degradation, recent evidence indicates that SUMO chains act as regulators of chromatin dynamics and genome stability by affecting the composition and assembly of DNA repair complexes on chromatin during the replication stress response [63].

Beyond the function of SUMOylation in regulating DNA repair and replication factors’ activity and cellular localization, SUMO metabolism is critical to the mobility of DNA lesions and their anchorage to NPC.

## 4. NPCs Anchor DNA Lesions in a SUMO-Dependent Manner to Promote DNA Repair

The double-layered nuclear membrane is penetrated by large macromolecular structures called NPCs that have an estimated mass of ~50 MDa in yeast and 112 MDa in vertebrates [84]. Cryo-electron microscopy has shown that the architecture of NPCs is highly conserved among eukaryotes [85]. Each NPC is assembled from multiple copies of ~30 different nucleoporins, which are called nucleoproteins or Nups. These proteins associate in distinct sub-complexes joined to each other, including eight cytoplasmic filaments, the symmetric central scaffold, and eight nucleoplasmic filaments, forming the nuclear basket [86]. The central scaffold is composed of an inner-ring complex surrounded by the outer rings containing cytoplasmic and nuclear domains. The inner ring constitutes a central channel abundant in FG-nucleoporins that facilitate the selective nucleocytoplasmic transport of molecules. The major building blocks of the outer rings are the Y-shaped Nup107-160 complexes (in humans and *S. pombe*), known as the Nup84 complex in budding yeast [86,87,88].

Beyond the canonical function of NPCs in the selective import/export of proteins and RNAs, those large structures contribute to the regulation of gene expression, 3D organization of the genomes, DNA repair processes, and maintenance of genome integrity [89,90,91]. Several studies have demonstrated that NPCs are an integral part of the DNA damage response (DDR), acting by promoting the transport of DNA repair factors by anchoring DNA lesions and by engaging alternative DNA repair pathways. Mutations in the Y-complex or the nuclear basket make yeast cells highly vulnerable to DNA damage and replication stress [74,91,92], although it is often not clear if this is a consequence of defective macromolecular transport or related to a direct function of NPCs in DNA repair. For example, the depletion of the human nuclear basket nucleoporin NUP153 leads to a defective import of the DDR mediator 53BP1 into the nucleus, resulting in an increased level of intrinsic replication stress and to cellular sensitivity to replication-blocking agents [93,94]. In buddying yeast, mutations in several nucleoporins of the Nup84 complex (e.g., *nup84*Δ or *nup133*Δ) lead to sensitivity to genotoxic drugs and replication stress [20,92]. Additionally, disruption of *NUP84* was reported to cause a delay in replication fork progression in the presence of DNA damage [95]. In fission yeast, the lack of Nup132 (*NUP133* in budding yeast and humans) leads to sensitivity to replication stress but not to DSBs or UV-induced DNA damage, and Nup132 promotes DNA replication recovery upon transient fork stalling [33].

Evidence gathered over the past two decades from numerous studies support the concept that NPCs act as docking sites for different types of DNA lesions. However, the exact NPC components involved in anchoring DNA lesions are unknown. The anchorage of DNA lesions is dependent on SUMO metabolism and both monoSUMOylation and SUMO chains’ formation, indicating that SUMO constitutes the key signal for NPC anchorage (Table 2). A current model from a budding yeast study indicates that the STUbL factor Slx8 associates with the Y complex of the NPC, providing a physical link between SUMOylated proteins at DNA damage sites and the NPC [20]. The NPC anchorage of persistent DSBs, heterochromatic DSBs, and eroded telomeres in several organisms requires Slx8 [21,31,38,96]. SUMO interaction motifs (SIMs) of STUbL would allow bridging SUMOylated repair factors to NPCs [97]. However, there is no structural information regarding Slx8-NPC interactions to improve our understanding of the anchorage function of the NPC. In addition, it remains unknown whether this interaction is conserved in other eukaryotes, and/or additional mechanisms of anchorage do exist. Indeed, several Nups contain SIM domains that may be instrumental to anchor SUMOylated DNA repair factors to NPCs. The NPC anchorage of DSBs is necessary to maintain genome integrity, but the mechanisms engaged at NPCs remain not entirely uncovered. Studies from different model organisms support the concept that DSBs at repeated sequences and/or heterochromatin are subjected to SUMOylation events. This is necessary for DSBs to shift away from their compartments and to spatially regulate the subsequent steps of DSB repair by HR. In *Drosophila*, heterochromatic DSBs relocate to the nuclear periphery in an Nse2- and PIAS-dependent manner [21]. In this system, SUMOylation inhibits the loading of RAD51 before relocation. At the periphery, STUbL stabilizes the interaction with repair sites and promotes the loading of Rad51, but how this step is achieved is currently unknown.

In budding yeast, SUMOylation plays a key role in the nucleolar dynamics by ensuring the compartmentalization of HR activities. Indeed, replication-born DSBs within rDNA sequences shift from the nucleolus to anchor to NPCs and maintain repeat integrity [23]. Recently, the mobility of individual rDNA repeats out of the nucleolus was shown to be dependent on the SUMOylation of factors that tether rDNA units to the nuclear periphery [98]. Moreover, the SUMOylation of the HR factor Rad52 enables its exclusion from the nucleolus, thus limiting deleterious recombination events within rDNA [99]. Preventing Rad52 SUMOylation leads to the formation of Rad52 foci inside the nucleolus and rDNA hyper-recombination. In budding yeast, eroded telomeres (i.e., in the absence of telomerase) undergo polySUMOylation and anchor to NPCs to facilitate the maintenance of telomeres lengths through a BIR type of repair, generating type II survivors [30,31,100]. PolySUMOylated telomeres are targeted to NPCs by the Slx5-Slx8 STUbL, and after anchorage to NPCs, they undergo deSUMOylation by the Ulp1 SUMO protease located at nuclear basket to unlock a Rad51-independent pathway [31]. Together, these pioneering studies point out how SUMOylation coordinates the nuclear positioning of DSBs and HR activities to maintain genome stability.

A recent study on budding yeast highlighted SUMO-independent alternative mechanisms by which NPCs regulate HR activity (Table 2). Replication forks that stall at telomere repeats relocate and anchor to the NPC, via the nucleoporin Nup1 of the nuclear basket, to promote a conservative HR type of repair [34]. In contrast, when anchorage to NPCs was prevented, stalled forks were subjected to a Rad51-dependent type of HR, leading to error-prone sister chromatid recombination (SCR) to maintain telomere length and bypass replicative senescence. Interestingly, Siz1- or Siz2-dependent SUMOylation was not required to promote this last pathway. Instead, it was proposed that Nup1 prevents SCR by regulating karyopherin functions in escorting cargo to the site of the stalled fork to channel their repair toward a conservative restart pathway.

Both chromatin context and repeated sequences affect the scenario by which DSBs are repaired; SUMOylation events appear necessary to move DSBs away from their compartments and prevent RAD51 loading until DSBs are relocated in a “safer” environment to complete DNA repair. Both similar and distinct scenarios have emerged for the repair of arrested forks.

## 5. SUMO-Based Regulation of Nuclear Positioning Regulates Replication Fork Repair

Akin to SUMO’s role as a nuclear positioning signal for DSBs, SUMO-dependent relocation of replication stress sites was recently discovered. In budding yeast, tri-nucleotide repeats, such as CAG, have a tendency to form secondary DNA structures prone to stall replication forks [101]. Such stalled forks relocate and anchor to NPCs in an Mms21-dependent SUMOylation and Slx5-dependent manner in late S-phase (Figure 2 and Table 2) [32]. Mutations of SIM domains of Slx5 were sufficient to prevent relocation, further supporting the role of STUbL in bridging SUMOylation and NPC anchorage [91,97]. This anchorage requires stalled forks to be processed by the end resection machinery to expose the ssDNA on which RPA is loaded. NPC anchorage requires the SUMOylation of RPA and the HR factors Rad52 and Rad59. As suggested at DSBs, SUMOylation and especially SUMO-RPA, which is known to interact with Slx8-Slx5 [31,102], prevent Rad51 loading before NPC anchorage. Indeed, Rad51 foci formation at stalled forks occurs only after relocation and anchorage [97], suggesting that not yet identified mechanisms are at work in the NPC to promote Rad51 engagement at forks stalled within repeated sequences. The relocation event is crucial for maintaining genome stability, as the lack of NPC anchorage leads to increased chromosomal fragility of CAG tracks [32], indicating that NPCs allow the engagement of specific mechanisms to maintain fork integrity. Thus, as observed for DSBs within repeated sequences, SUMOylation restrains Rad51-dependent HR events that can be detrimental when forks are arrested at repeated sequences. This routing of repeats-induced stalled forks toward NPCs could allow an error-free and Rad51-dependent fork restart pathway.

In fission yeast, forks arrested at the *RTS1*-RFB, which mediates a DNA-bound protein block to replisomes, were recently shown to relocate and anchor to NPCs in S-phase (Figure 2 and Table 2) [33]. This anchorage event requires the E3 SUMO ligase Pli1 and the Slx8 STUbL pathway, indicating that SUMOylation is the key nuclear positioning signal, but the exact SUMOylated targets are unknown. However, the underlying type of SUMOylation is the SUMO chain. Indeed, abrogating the formation of SUMO chains by mutating all acceptor lysine to arginine in *SUMO-KallR* mutant leads to a lack of relocation to the nuclear periphery. Moreover, anchorage to the NPC requires Rad51 binding to arrested forks, as well as its strand exchange activity, suggesting that arrested forks need to be remodeled by HR activity to be prone to anchorage. One possibility is that Rad51-dependent recombination/replication DNA structures trigger the recruitment of specific factors subjected to the SUMO chain formation critical to NPC anchorage. These data indicate that, contrary to the repeats-induced stalled forks, Rad51 binding occurs before anchorage to NPCs.

Although SUMO chains signal relocation, they negatively impact the efficiency of RDR. Indeed, the efficiency of RDR was increased in the absence of SUMO chains. Destabilizing the interaction between SUMO and the E2 conjugating enzyme Ubc9 (in the SUMO-D81R mutant) also stimulated the efficiency of RDR. NPC anchorage is then necessary to clear off SUMO conjugates by the proteasome and the SENP protease Ulp1, two activities enriched at the nuclear periphery. In the absence of Nup132, Ulp1 is delocalized from NPC and less expressed [103]. In this genetic context, arrested forks were properly anchored to NPCs but RDR efficiency was decreased, revealing a novel post-anchoring function of NPCs in ensuring replication restart. The artificial tethering of Ulp1 to the RFB was sufficient to restore RDR efficiency. These data suggest the existence of at least two spatially segregated RDR pathways whose choice is under SUMO control. Pli1 would be recruited early at arrested forks to safeguard fork integrity by limiting the degradation of the nascent strand. SUMO chains, arising as a presumable consequence of Pli1 activity, may restrain a type of DNA synthesis for replication resumption, creating a commitment to NPCs anchorage to overcome the SUMO chain’s inhibitory effect. When only monoSUMOylation occurs, the arrested forks remain in the cytoplasm, and the fork restart occurs efficiently. During the HR-mediated fork restart, the DNA polymerase delta synthetizes both strands of the restarted fork, in contrast to origin-born replication fork [42]. Whether SUMOylation events at the RFB influence the use of distinct DNA polymerases during fork restart is unknown. Interestingly, the defective STUBL pathway resulted in a marked increase in the mobility of the RFB, whereas the absence of Pli1 resulted in a global decrease in RFB’s mobility, suggesting that the level of SUMOylation at sites of replication stress is critical for nuclear movement. Interestingly, the formation of liquid-like repair centers of Rad52, a SUMO target, requires the correct assembly of intracellular microtubule filaments in budding yeast [104]. It is unknown, however, if an interplay between nuclear filaments and SUMO metabolism exists and impacts the processing of DNA lesions.

Overall, these finding reveal that the switch between monoSUMOylation and the SUMO chain formation at arrested replisomes likely constitutes a quality-control step that dictates the choice of replication fork repair pathways in the nuclear space. Moreover, the SUMO metabolism differentially influences the fate of arrested replisomes according to sequences’ context; at repeated sequences, SUMOylation restrains Rad51 loading until the stalled forks anchor to NPCs, whereas at unique sequences, SUMOylation is necessary to maintain fork integrity until SUMO chains trigger NPC anchorage to allow an efficient fork restart. This suggests that additional features such as chromatin landscape influences SUMOylation features at sites of replication stress.

## 6. Concluding Remarks

SUMOylation connects replication stress sites to NPCs that act as molecular hubs to regulate HR activity. Different scenarios have emerged according to the type of fork obstacles and their surrounding sequences environment and organisms. The mechanisms triggering the relocation of forks arrested at repeated sequences, such as at expanded CAG, are presumably distinct from those involved in the relocation of protein-mediated fork stalling. It is evident that cells have evolved pathways to restrict the access of Rad51 at repeated sequences to limit deleterious HR events and preserve a constant size of repeats. Such pathways may limit fork-restart efficiency when the forks arrest at unique sequences. However, all the scenarios reveal that a spatially segregated SUMO metabolism is critical to ensure genome integrity at replication stress sites. Many questions remain to be addressed: How is the division of labor organized between distinct E3 SUMO ligases in yeast and human nuclei upon replication stress? What are the mechanisms engaged at the NPC or nuclear periphery that ensure an efficient and error-free fork restart? How are these NPC-related mechanisms coordinated with the global DDR response? How do chromatin organization and potential histone marks influence SUMO metabolism at sites of replication stress? Finally, most DSBs do not relocalize to the nuclear periphery or NPCs, raising questions about how the molecular and structural determinants make replication stress sites prone to relocation and NPC anchorage. Deciphering the repertoire of SUMOylated factors at replication forks upon various replication-blocking agents will certainly provide additional layers to answer these questions.

## Figures and Tables

**Figure 1 genes-12-02010-f001:**
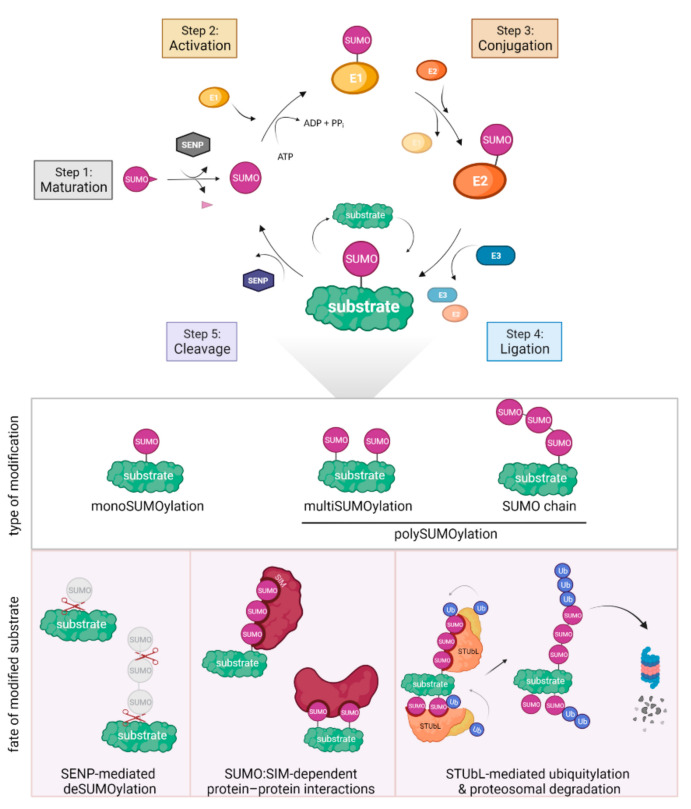
SUMO metabolism and functions. Top panel: cycle of SUMOylation. Bottom panel: function of the different types of SUMOylation.

**Figure 2 genes-12-02010-f002:**
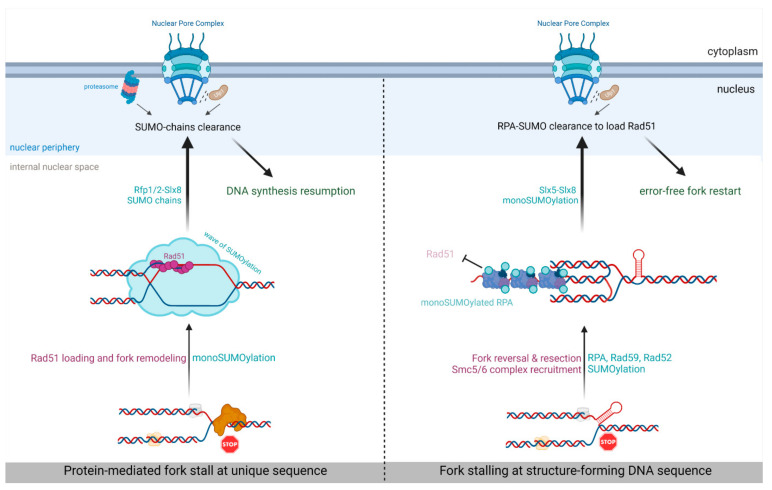
Routing towards NPCs for DNA-bound, protein-mediated fork arrest (**left** panel) and structure-forming-mediated fork stalling (**right** panel).

**Table 1 genes-12-02010-t001:** Players of the SUMO pathway in humans, *Saccharomyces cerevisiae* and *Schizosaccharomyces pombe*.

SUMO Pathway Component		Humans	*S. cerevisiae*	*S. pombe*
Small ubiquitin-like modifier (SUMO)		SUMO-1, SUMO-2, SUMO-3, SUMO-4, SUMO-5	Smt3	Pmt3
Activating enzyme (E1)		SAE1 SAE2	Aos1 Uba2	Rad31 Fub2
Conjugating enzyme (E2)		Ubc9	Ubc9	Hus5
SUMO ligase (E3)	SP-RING type	PIAS1, PIAS2, PIAS3, PIAS4 Mms21	Siz1, Siz2 Mms21 Zip3	Pli1Nse2
	other	RanBP2 * [69] HDAC4 [70], KPA1 [71], Pc2 [72], Topors [73]		
SUMO-targeted ubiquitin ligase (STUbL)		RNF4 RNF11	Slx5-Slx8 Uls1	Rfp1/Rfp2-Slx8 Rrp2 (predicted)
Sentrin/SUMO-specific protease (SENP)		SENP1 °^,^*, SENP2 °^,^*, SENP3, SENP5 ° SENP6, SENP7	Ulp1 °^,^* Ulp2	Ulp1 °^,^* Ulp2

* Localized at the nuclear pore complex. ° Involved in SUMO maturation.

**Table 2 genes-12-02010-t002:** Comparison of systems of replication stress relocation to NPC/nuclear periphery.

Type of Obstacle	Protein-Mediated Fork Arrest	Structure-Forming DNA Sequence	Telomere-Specific Replication Stress		Aphidicolin Induced Replication Stress
System description	Site-specific RFB blocking a single replisome in a polar manner	Expanded trinucleotide repeats forming hairpin structures that stall replisomes	Stalled replisomes at telomere repeats in telomerase-negative cells	Telomere-specific replication stress induced by POT1 dysfunctions	Global replication fork stalling induced
Organism	*S. pombe*	*S. cerevisiae*	*S. cerevisiae*	human cell lines	human cell lines
Relocation and anchorage requirements	● Rad51-dependent fork remodeling ● Pli1 ● SUMO chain ● Rfp1-Slx8, ● Rfp2-Slx8 ● NPC-anchorage site unknown	● Nascent DNA degradation (by Mre11, Exo1, Dna2) ● Mms21 ● SUMOylation of RPA, Rad52, Rad59 ● Slx5-SUMO interaction ● Nup1, Nup84	● Nup1	● F-actin polymerization ● ATR pathway ● Nup62, Nup153, TPR	● F-actin polymerization
Relocation outcomes	Ulp1-NPCs alleviate inhibitory effect of SUMO chains on HR-mediated fork restart	Rad51 loading to promote error free fork restart and preventing CAG repeat instability	Promoting conservative fork restart pathway to avoid error-prone Rad51-dependent SCR	Preventing SCR at telomeres to promote the maintenance of repetitive DNA	Promoting replication stress response to ensure fork restart and prevent mitotic abnormalities.
Reference	[33]	[32,34,97]	[34]	[36]	[35]

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
