# Peer review of "SUMO-Based Regulation of Nuclear Positioning to Spatially Regulate Homologous Recombination Activities at Replication Stress Sites"

_genes, 2021, doi:10.3390/genes12122010_

Round 1
Reviewer 1 Report
The role of SUMOylation in DNA repair, the targeting of damaged DNA in particular, is a very interesting topic. The review by Schirmeisen et.al is well compiled with the latest updates on this topic that will be interesting to the field. I only have a couple of minor issues listed below.
- line 25, should it be 2X10^16 instead.
- line 56, BIR is only one of the HR mechanisms in fork repair and its function is limited to the proximity of chromosomal ends. A major function of HR in fork recovery is via template switching.
- line 131-134, the authors mentioned pioneering studies in yeasts....while giving a model of human CtIP might be a bit confusing.
Author Response
The role of SUMOylation in DNA repair, the targeting of damaged DNA in particular, is a very interesting topic. The review by Schirmeisen et.al is well compiled with the latest updates on this topic that will be interesting to the field. I only have a couple of minor issues listed below.
We would like to thank the Reviewer for comments on our manuscript. Certain issues raised by the Reviewer were corrected and listed below.
1. line 25, should it be 2X10^16 instead.
Corrected.
2. line 56, BIR is only one of the HR mechanisms in fork repair and its function is limited to the proximity of chromosomal ends. A major function of HR in fork recovery is via template switching.
We agree that HR catalyses multiple transactions at stressed forks, according to the initial causes of replication blockade. Template switching is now included within text as a potent mechanism of Rad51 activity at dysfunctional forks (line 58).
3. line 131-134, the authors mentioned pioneering studies in yeasts... while giving a model of human CtIP might be a bit confusing.
Line 131-134 were complemented by the following sentence to avoid confusion: “Furthermore, studies conducted in higher eukaryotes, also allowed to describe numerous SUMO-targets among DNA repair proteins.” (line 135-137). We hope that this modifications will satisfy the Reviewer.
Reviewer 2 Report
I have to admit that all I knew about sumoylation was that it was a post translational modification. In fact, the review has driven my attention and interest towards the role of sumoylation in regulating DNA repair, replication factors activity and subcellular localization as well as in anchoring certain lesioned DNA to the nuclear pore complexes.
The review is well-written. To my best knowledge, the manuscript is clear, relevant for the field and presented in a well-structured manner. The cited references are appropriate and up-to-date. The tables and schemes are easy to interpret and understand.
I will make some comments, anyway, just in case authors want to take them into account to widen the public that would benefit from reading the review.
- Is there an operational definition that allows to distinguish in an experimental situation whether you have break-induced replication (BIR), recombination-dependent replication (RDR) or other homologous recombination subpathways such as double holliday junction subpathway? (see doi:10.1016/j.tig.2018.04.002).
- In order to know what sort of evidence sustained some key points, I had to go and read the original works. I wonder if it would be feasible to include short explanations to stress the evidence weight for the main assertions. This happened to me in relation to assertions such as that forks inhibited by DNA pol or stalled at telomeres can undergo displacement and association with nuclear pore complexes (NPCs), or that NPCs affect the DDR (either directly or due to the alteration of cytoplasmic-nuclear transport, (citations 35,36,37; 74,91,92; 21).
……………….Specific comments………………………………………………………….
- Akin to ubiquitin….monosumoylation (suggests that ubiquitination is always mono..)
- Line 346: substitute restrains by restrain
- Line 366: substitute suggest by suggests
- Line 381: substitute:
How these NPC-related mechanisms are coordinated …? by
How are these NPC-related mechanisms coordinated…?
- Line 382. Substitute:
How chromatin organization and potential histone marks influences…? by
How do chromatin organization and potential histone marks influence…?
……………………………………………………………………………………………………….
Author Response
I have to admit that all I knew about sumoylation was that it was a post translational modification. In fact, the review has driven my attention and interest towards the role of sumoylation in regulating DNA repair, replication factors activity and subcellular localization as well as in anchoring certain lesioned DNA to the nuclear pore complexes.
The review is well-written. To my best knowledge, the manuscript is clear, relevant for the field and presented in a well-structured manner. The cited references are appropriate and up-to-date. The tables and schemes are easy to interpret and understand.
I will make some comments, anyway, just in case authors want to take them into account to widen the public that would benefit from reading the review.
We would like to thank the Reviewer for comments on our manuscript. Certain issues raised by the Reviewer are discussed and listed below.
1. Is there an operational definition that allows to distinguish in an experimental situation whether you have break-induced replication (BIR), recombination-dependent replication (RDR) or other homologous recombination subpathways such as double holliday junction subpathway? (see doi:10.1016/j.tig.2018.04.002).
We thank Reviewer for pointing out the work by Kramara et al about BIR in eukaryotic organisms. To address the point - direct analysis of replication profiles by 2DGE allow to examine whether replication block is connected with a DSB or HJ formation. BIR is resulting from one-ended DSB at replication fork. Of course, it is not easy to study those events, however, by employing yeast genetics and designing particular replication blocks or site-specific DSBs it is possible to have an insight into mechanisms of repair of DSBs in heterochromatin, eroded telomeres, stalled forks within secondary DNA structures or tight protein blocks. In S. pombe model it was shown that tight protein mediated replication fork stall is suitable to turn on relocation signalling without DSB formation and is not transformed into one-ended DSB. However, we think that providing additional details about the experimental conditions to distinguish between BIR, RDR or other HR sub-pathways is out of the scope of this review.
2. In order to know what sort of evidence sustained some key points, I had to go and read the original works. I wonder if it would be feasible to include short explanations to stress the evidence weight for the main assertions. This happened to me in relation to assertions such as that forks inhibited by DNA pol or stalled at telomeres can undergo displacement and association with nuclear pore complexes (NPCs), or that NPCs affect the DDR (either directly or due to the alteration of cytoplasmic-nuclear transport, (citations 35,36,37; 74,91,92; 21).
We did our best to address this point. The sentence line 87 was modified as follows: “In human cells, fork stalled upon inhibition of DNA polymerases exhibit relocation to the nuclear periphery and replication stress at telomeres lead to telomeres association with NPCs”. We also added the REF 20 to the following sentence: Budding yeast Ulp1 is localized at the nuclear periphery through interactions with the Y-complex of the NPC (Nup84) and the nuclear basket (Nup60–Mlp1/2), whereas Ulp2 is located in the nucleoplasm”. Regarding the references 91 and 92, we modified the sentence as follows: ”Mutations in the Y-complex or the nuclear basket make yeast cells highly vulnerable to DNA damage and replication stress” (line 228). We hope that these modifications will satisfy the Reviewer.
Specific comments
3. Akin to ubiquitin …. monosumoylation (suggests that ubiquitination is always mono..)
‘Akin to ubiquitin’ phrase was replaced within the text to avoid confusion (line 122 to line 116).
4. Line 346: substitute restrains by restrain
Corrected.
5. Line 366: substitute suggest by suggests
Corrected.
6. Line 381: substitute:
How these NPC-related mechanisms are coordinated …? by
How are these NPC-related mechanisms coordinated…?
Corrected.
7. Line 382. Substitute:
How chromatin organization and potential histone marks influences…? by
How do chromatin organization and potential histone marks influence…?
Corrected.
Reviewer 3 Report
The article reviews the role of sumoylation in DNA repair. The review is comprehensive and written well for the lay reader.
I have only one minor comment:
The phrase "membrane-less compartmentalized nucleus" is confusing. The nucleus has a membrane. I presume the authors allude to nuclear compartments being membraneless - this needs to be rephrased to avoid confusion.
Author Response
The article reviews the role of sumoylation in DNA repair. The review is comprehensive and written well for the lay reader.
We thank the Reviewer for reading our manuscript. The confusing ‘membrane-less’ wording was removed from the text (line 70). We hope that in current version it satisfies the Reviewer.
I have only one minor comment:
The phrase "membrane-less compartmentalized nucleus" is confusing. The nucleus has a membrane. I presume the authors allude to nuclear compartments being membraneless - this needs to be rephrased to avoid confusion.
This sentence has been modified as follow: “Eukaryotic genomes are 3D folded in a highly compartmentalized nucleus that has distinct chromatin environment and DNA repair capacity”.